# Uveal melanoma: Long-term survival

**Tomas Radivoyevitch**[1], **Emily C. Zabor**[1], **Arun D. Singh**[2]*

**1** Quantitative Health Sciences, Lerner Research Institute, Cleveland Clinic, Cleveland, Ohio, United States of America, **2** Ophthalmic Oncology, Cole Eye Institute, Cleveland Clinic, Cleveland, Ohio, United States of America

\* SINGHA@ccf.org

## Abstract

### Purpose

The long-term survival of uveal melanoma patients in the US is not known. We compared long-term survival estimates using relative survival, excess absolute risk (EAR), Kaplan-Meier (KM), and competing risk analyses.

### Setting

Population based cohort study.

### Study population

Pooled databases from Surveillance, Epidemiology, and End Results data (SEER, SEER-9 +SEER-13+SEER-18).

### Main outcome measure

Overall Survival (OS), Metastasis Free Survival (MFS) and relative survival, computed directly or estimated via a model fitted to excess mortality.

### Results

There were 10678 cases of uveal melanoma spanning a period of 42 years (1975–2016). The median age at diagnosis was 63 years (range 3–99). Over half the patients were still alive at the end of 2016 (53%, 5625). The KM estimates of MFS were 0.729 (0.719, 0.74), 0.648 (0.633, 0.663), and 0.616 (0.596, 0.636) at 10, 20, and 30 years, respectively. The cumulative probabilities of melanoma metastatic death at 10, 20 and 30 years were 0.241 (0.236, 0.245), 0.289 (0.283, 0.294), and 0.301 (0.295, 0.307). In the first 5 years since diagnosis of uveal melanoma, the proportion of deaths attributable to uveal melanoma were 1.3 with rapid fall after 10 years. Death due to melanoma were rare beyond 20 years. Relative survival (RS) plateaued to ~60% across 20 to 30 years. EAR parametric modeling yielded a survival probability of 57%.

**Data Availability Statement:** The full data set is publicly available via the NCI's SEER program upon signing a Data Use Agreement with the NCI (https://seer.cancer.gov/data/access.html). Anyone accessing this data at the individual level must sign this agreement. Grouped data used in the EAR

Poisson regressions of Fig 2E are available via GitHub (https://github.com/radivot/SEERaBomb).

**Funding:** There is no funding associated with this work.

**Competing interests:** We have no competing interests.

## Conclusions

Relative survival methods can be used to estimate long term survival of uveal melanoma patients without knowing the exact cause of death. RS and EAR provide more realistic estimates as they compare the survival to that of a normal matched population. Death due to melanoma were rare beyond 20 years with normal life expectancy reached at 25 years after primary therapy.

## Introduction

There is a paucity of uveal melanoma survival data beyond 15 years. A multicenter collaborative effort to develop American Joint Commission on Cancer (AJCC) staging for uveal melanoma reported that 19% of patients had died due to metastatic melanoma at 10 years [1]. The Collaborative Ocular Melanoma Study (COMS) reported 20% died of melanoma (10% vs 30% for smaller vs larger sizes in younger vs older patients) at 12 years [2], and two studies reported 40% died of melanoma at 10 years [3, 4]. Beyond 15 years following ocular treatment, cohort thinning presents challenges [5–7]. Large population-based datasets with long-term follow up are needed to assess long-term survival in uveal melanoma patients [8, 9]. In a Danish study of 302 patients diagnosed in 1943–1952, 50% died of uveal melanoma at 25 years [8]. In a Finnish study of 289 patients diagnosed in 1962–1981, 52% died of uveal melanoma at 35 years [9]. Here we add new US estimates based on Surveillance, Epidemiology, and End Results (SEER) cases diagnosed in 1975–2016.

Critical to estimating cancer-specific long-term death rates is how the presence of metastasis is ascertained. Hospital registries [5, 7] and national databases can be inaccurate [9]. As such, COMS designed its own system of documenting metastasis [10]. In relative survival (RS) [11] approaches, survival probabilities observed in those with uveal melanoma are divided by survival probabilities expected in age-year-sex-matched simulated "normal" individuals from the same population. RS avoids use of cause of death (COD) and uses background mortalities instead. In a hazard focused approach, person-years at risk of death (PY) in the same age-year-sex bin are tallied across patients and multiplied into background mortality rates to form, for each arbitrarily chosen time interval after diagnosis, expected numbers of deaths E [12]. Such E, with total PY and observed deaths (O) in the same intervals, yield time courses of relative risks (RR = O/E) and excess absolute risks (EAR = (O-E)/PY) of mortality. In this framework, an RR of 2 implies life-expectancy halving and an EAR(t) of 0.05 implies a death rate increase of 5% per year at time t.

SEER cancer registry data [13] meet the prerequisites of long-term analyses, such as large numbers of patients [14] and long follow-up [15]. It is the source of summary statistics that are published annually as the national cancer report in the United States [16]. We use SEER cancer registry data here to provide a US-based estimate of uveal melanoma lifetime mortality. We compare estimates obtained using EAR methods to those obtained using RS and COD-based methods.

## Methods

### Data selection

Since SEER data are deidentified and publicly available, IRB approval was not necessary to conduct this study. SEER database ASCII files that include treatment information, provided in SEER_1975_2016_CUSTOM_TEXTDATA.d04122019.zip, were downloaded on April 16[th],

2019 and processed and analyzed using the R package SEERaBomb [17]. This software merges all three SEER databases (SEER-9+SEER-13+SEER-18) to provide more cases than can be retrieved conventionally using SEER*stat. It also accesses background US mortality rates in the Human Mortality Database (https://www.mortality.org/). Uveal melanoma cases were identified using International Classification of Disease for Oncology (ICD-O-3) morphology (melanoma: 8,720–8,790) and site [C69.3 (choroid), C69.4 (ciliary body and iris), and C69.2 (retina)] codes [14]. Diagnoses occurred between 1/1/1975 and 12/31/2016. R scripts that produced our results are provided (**S1 Text**).

## Data analysis

The Kaplan-Meier (KM) method [18] in the R package survival was used to form estimates of overall survival (OS) and metastasis free survival (MFS) [19, 20]. Both were calculated from dates of diagnoses. For MFS, times to metastasis are approximated as times of deaths due to the uveal melanoma and deaths due to other causes were treated as right-censored. Cumulative incidence (CI) of mortality [21] was computed using the competing risks function cuminc in the R package cmprsk [22], in which deaths due to uveal melanoma were the events of interest and deaths due to other causes were treated as a competing event. RS [23] denominator survival probabilities were computed using the R function simSurv in the R package SEERaBomb [17]. This function uses Human Mortality Database US mortality rates to generate the probability of survival in a comparable (same age-sex-year) US cohort. It accomplishes this by simulating each individual forward in time from age at diagnosis, incrementing year and age one year at a time and using updated morality rates at each step (i.e. age and year both increase by one year at each step). If a death arises in a step, the time of it within the year is selected from a uniform distribution over that year. For each uveal melanoma case, three normal counterparts were simulated to form the control cohort. Relative survival was then calculated as observed uveal melanoma OS divided by the simulated expected OS. EAR estimates were computed using the SEERaBomb R function mortality since diagnosis (msd).

## Results

### Study demographics

There were 10678 cases of uveal melanoma spanning a period of 42 years (1975–2016).

The median age at diagnosis was 63 years (range 3–99). The vast majority of cases were race-classified as white (97%). Over half the patients were alive (5625) as of 12/31/2016. Of 5053 cases that died by this date, the cause of death was uveal melanoma in 2266 (44.8%) (**Table 1**).

Greater proportions of deaths were attributed to causes other than uveal melanoma metastasis beyond 10 years (77%–94%) (**Fig 1**).

In 2000, SEER expanded from 13 to 18 registries. As a result, there were disproportionately more person-years (PY) at risk of death 0–16 years after diagnoses, wherein risks of death by melanoma are greater. Focusing on SEER-9 diagnoses in 1975–1986, 37% [399/(399+678)] were due to uveal melanoma. Assuming 163 still alive at the end of 2016 will die of other causes, the proportion falls to 32%. This raw proportion is our <u>first estimate</u> of the lifetime risk of death due to uveal melanoma.

### Survival metrics

**Kaplan Meier (KM) analysis.** The KM estimates of survival probabilities at 10, 20, and 30 years were 0.529 (95% confidence interval (CI): 0.518, 0.541), 0.305 (95% CI: 0.291, 0.319), and

**Table 1. Uveal melanoma.** Demographic distributions (n = 10678).

| ASPECT | Group | Number of cases (%) |
|---|---|---|
| Year of Diagnosis | 1973–1982 | 864 (8.1) |
| | 1983–1992 | 1014 (9.5) |
| | 1993–2002 | 2302 (21.6) |
| | 2003–2016 | 6498 (60.9) |
| Gender | Male | 5580 (52.3) |
| | Female | 5098 (47.7) |
| Age at Diagnosis | Range | 3–99 |
| | Q1-Q3 | 52–72 |
| | Median | 63 |
| Laterality | Right Eye | 5266 (49.3) |
| | Left Eye | 5317 (49.8) |
| | Bilateral | 2 (0.1) |
| | Unknown | 93 (0.8) |
| Race | White | 10347 (96.9) |
| | Black | 80 (0.75) |
| | Other / Unknown | 251 (2.4) |
| Method of treatment | Radiation only | 6013 (56.3) |
| | Surgery Only | 4440 (41.6) |
| | Combination | 194 (1.8) |
| | Other/ Not specified | 31 (0.3) |
| Status/ Metastasis | Alive | 5625* |
| | Dead with metastasis | 2266* |
| | Dead with other causes | 2787* |

*Percentages of these numbers are omitted intentionally as they could be misleading.

0.181 (95% CI: 0.166, 0.198) for OS and 0.729 (95% CI: 0.719, 0.74), 0.648 (95% CI: 0.633, 0.663), and 0.616 (95% CI: 0.596, 0.636) for MFS, respectively (**Fig 2A**).

OS was worse than MFS at all time points, as expected, as the assumption under MFS is that a patient can only die of uveal melanoma. As MFS flattens after 30 years, MFS = 0.606 (95% CI: 0.584, 0.629) at 35 years, and by calculating 1-MFS we obtain our second estimate of life-time risk of death due uveal melanoma of 39% (95% CI: 37.1%, 41.6%) (**Table 2**).

**Competing risk analysis.** To properly control for competing risks by treating both death by other causes and death by uveal melanoma on equal footings, a standard R package for competing risk analyses was used. This yielded cumulative probabilities of death at 10, 20 and 30 years of 0.228 (95% CI: 0.224, 0.233), 0.404 (95% CI: 0.397, 0.411), and 0.516 (95% CI: 0.507, 0.525) for other causes of death and 0.241 (95% CI: 0.236, 0.245), 0.289 (95% CI: 0.283, 0.294), and 0.301 (95% CI: 0.295, 0.307) for death by metastatic uveal melanoma (**Table 2**). Thus, 30% (95% CI: 29%, 31%) is our third estimate of the lifetime risk of death due to uveal melanoma. Plots of these risks at all time points (**Fig 2B**) reveal that the risk of death from metastatic uveal melanoma is higher than other causes in the first 10 years since the ocular diagnosis/ treatment and lower thereafter as melanoma death risks begin to plateau but risks of death due to other causes continue to rise.

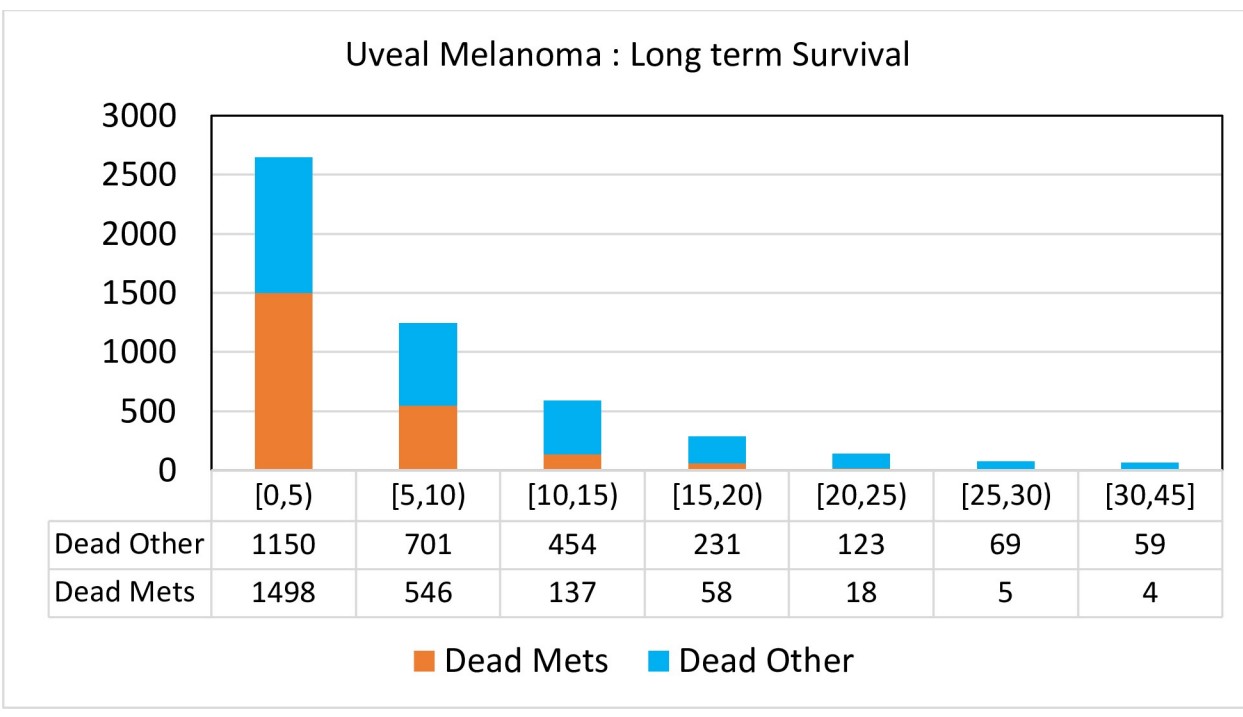

**Fig 1. In the first 5 years since diagnosis of uveal melanoma, the proportion of deaths attributable to uveal melanoma were 1.3 with rapid fall after 10 years.** Death due to melanoma were rare beyond 20 years.

**Relative survival.**    We first compare OS to its corresponding expected survival curve obtained by simulating three US age-sex-year matched normal "healthy" individuals for each age-sex-year at diagnosis of uveal melanoma (**Fig 2C**).

OS was worse in uveal melanoma patients than in the simulated normal population, which can be thought of as survival expected if the uveal melanoma patients did not actually have uveal melanoma. The gap between the curves narrows after approximately 25 years (**Fig 2C**). Relative survival (RS) is the ratio of the lower curve divided by the upper curve. It plateaus to ~60% across 20 to 30 years (**Fig 2D**).

Using 1-RS at 25 years yields a <u>fourth estimate</u> of death due to uveal melanoma of 40% (95% CI: 36%, 44%).

**Excess absolute risks.**    RS instability at large times due to cohort thinning and the unit of data being individuals is solved in the EAR approach by pooling PY at risk in time intervals across patients and increasing interval lengths as follow up times increase to boost numbers of deaths in late intervals. Modeling EAR as the sum of a smooth Gamma function wave and a triangle wave yields an EAR area under the curve (AUC) of 0.555 (**Fig 2E**) and thus a survival probability of $e^{-0.555} = 0.57$. Our <u>fifth estimate</u> of the probability of death due to uveal melanoma is thus 43%.

**Comparison of metrics.**    The lifetime risk of death due to uveal melanoma, estimated 5 different ways, ranged from 30%-43%. A comparison of curves underlying 4 of the estimates is presented in **Fig 2F**.

**Age dependence.**    Uveal melanoma mediated death risks are dominant for ~40 years in those younger than 50 years old at diagnosis (Fig 3A), but only for ~10 years in those ≥50 years old (Fig 3B). Lifetime risks of death by melanoma in these two age groups are ~30% (Fig 3C) and ~50% (Fig 3D), respectively, based on relative survival. Furthermore, in the younger

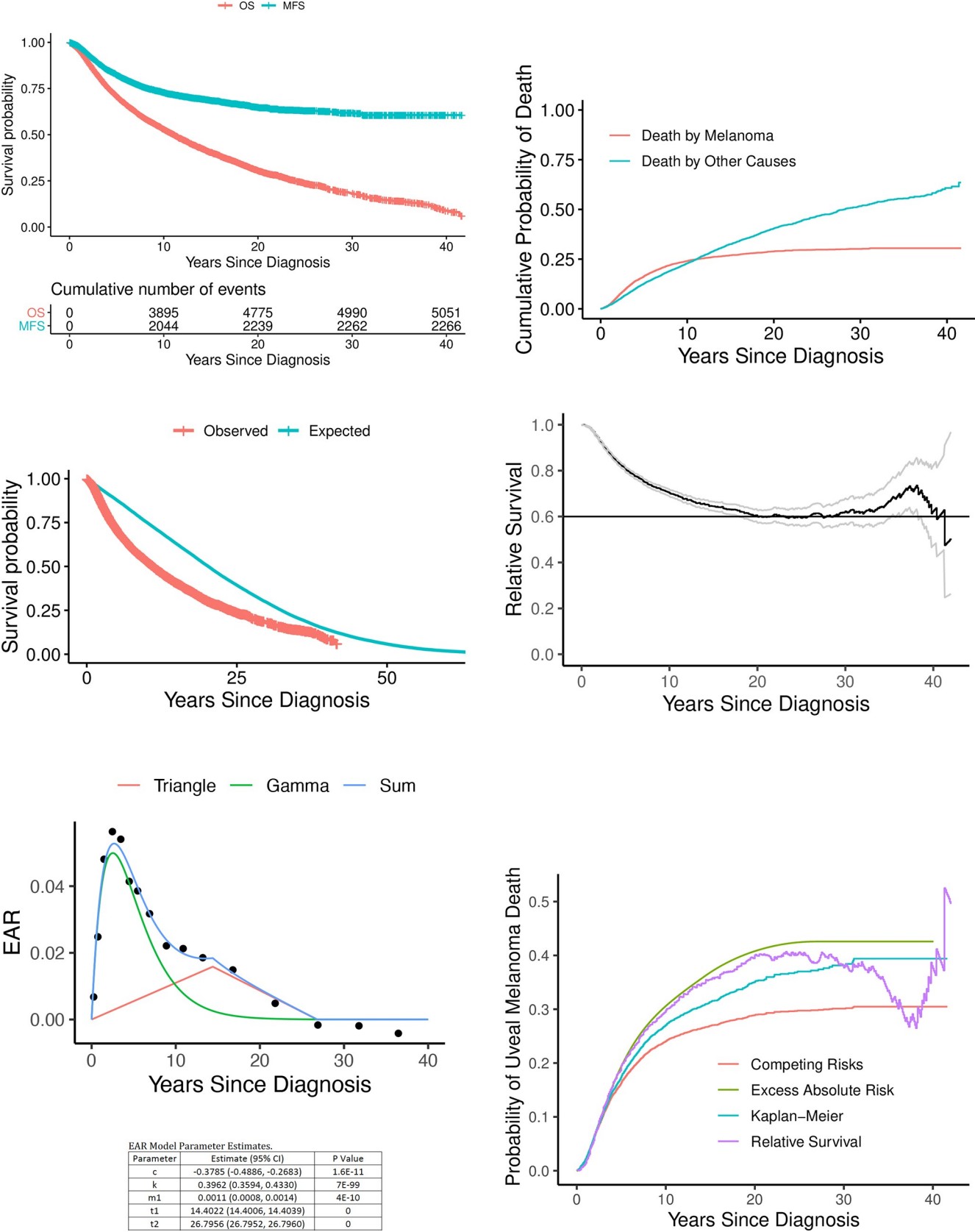

**Fig 2.** Kaplan-Meier plots (A). OS is difficult to interpret since everyone dies in the long run. MFS counts only deaths due to melanoma. Note that very few patients die due to melanoma 10 years after diagnosis. Cumulative incidence of death (B). These cumulative probabilities of death are integrals of death probability densities that account for competing risks. OS observed vs expected (C). Cumulative incidence of death. OS observed vs expected (C). Relative survival (D) is the ratio of the lower curve divided by the upper curve. The long-term mortality attributable to melanoma is estimated from the stable portion of this curve that lies between 20 and 30 years. (E) Trigam model fit to excess absolute risks of death due to uveal melanoma. Algebraic forms of the triangle wave and gamma function are provided above. Poisson regression was used to fit their sum to EAR "data" generated by the R function msd in SEERaBomb. This excess hazard adds to a subject's background hazard when diagnosed with uveal melanoma. (F). Comparison of survival methods. The EAR and the RS curves ignore cause of death. The other two curves rely on cause of death information. The KM-based 1-MFS curve is incorrect relative to the cumulative incidence (CI) curve because it does not account for competing risks.

age group (Fig 3E) relative to the older age group (Fig 3F), the peak EAR is lower and the triangle wave is not detectable. Negative EARs at large times in Fig 3F are possible, if additional surveillance results in better health care, but 95% confidence intervals of these estimates are wide and include 0, so we cannot claim this: the estimates of the last two EARs are -0.0188 (-0.0598, 0.0222) and -0.0241 (-0.113, 0.0648).

## Discussion

Determining long term mortalities attributable to uveal melanoma poses specific practical challenges that include a paucity of long-term data in institutional studies, the role of competing other causes of death that confound interpretations, and difficulty in ascertaining exact cause of death. Of the few published studies, most are based on nationwide populations

**Table 2. Long term survival estimates: Comparison of metrics (95%, CI).**

| Kaplan–Meier Estimates: Overall and Metastasis Free survival | | |
| --- | --- | --- |
| **Year** | **Overall Survival** | **Metastasis Free Survival** |
| 5 | 0.707 (0.697, 0.716) | 0.823 (0.815, 0.831) |
| 10 | 0.529 (0.518, 0.541) | 0.729 (0.719, 0.740) |
| 15 | 0.405 (0.393, 0.418) | 0.686 (0.674, 0.699) |
| 20 | 0.305 (0.291, 0.319) | 0.648 (0.633, 0.663) |
| 25 | 0.236 (0.222, 0.251) | 0.630 (0.613, 0.647) |
| 30 | 0.181 (0.166, 0.198) | 0.616 (0.596, 0.636) |
| 35 | 0.140 (0.124, 0.158) | 0.606 (0.584, 0.629) |
| **Cumulative Probabilities of Death** | | |
| | **Other Causes** | **Metastatic Deaths** |
| 5 | 0.126 (0.119, 0.133) | 0.163 (0.155, 0.170) |
| 10 | 0.228 (0.219, 0.238) | 0.241 (0.231, 0.250) |
| 15 | 0.325 (0.314, 0.337) | 0.269 (0.259, 0.279) |
| 20 | 0.404 (0.390, 0.417) | 0.289 (0.278, 0.300) |
| 25 | 0.466 (0.450, 0.482) | 0.297 (0.286, 0.309) |
| 30 | 0.516 (0.499, 0.533) | 0.301 (0.289, 0.313) |
| 35 | 0.555 (0.537, 0.574) | 0.305 (0.292, 0.317) |
| **Relative Survival Estimates** | | |
| | .. | **Relative Survival** |
| 5 | .. | 0.807 (0.797, 0.818) |
| 10 | .. | 0.700 (0.685, 0.715) |
| 15 | .. | 0.644 (0.625, 0.664) |
| 20 | .. | 0.601 (0.574, 0.629) |
| 25 | .. | 0.599 (0.563, 0.638) |
| 30 | .. | 0.612 (0.561, 0.668) |
| 35 | .. | 0.660 (0.586, 0.743) |

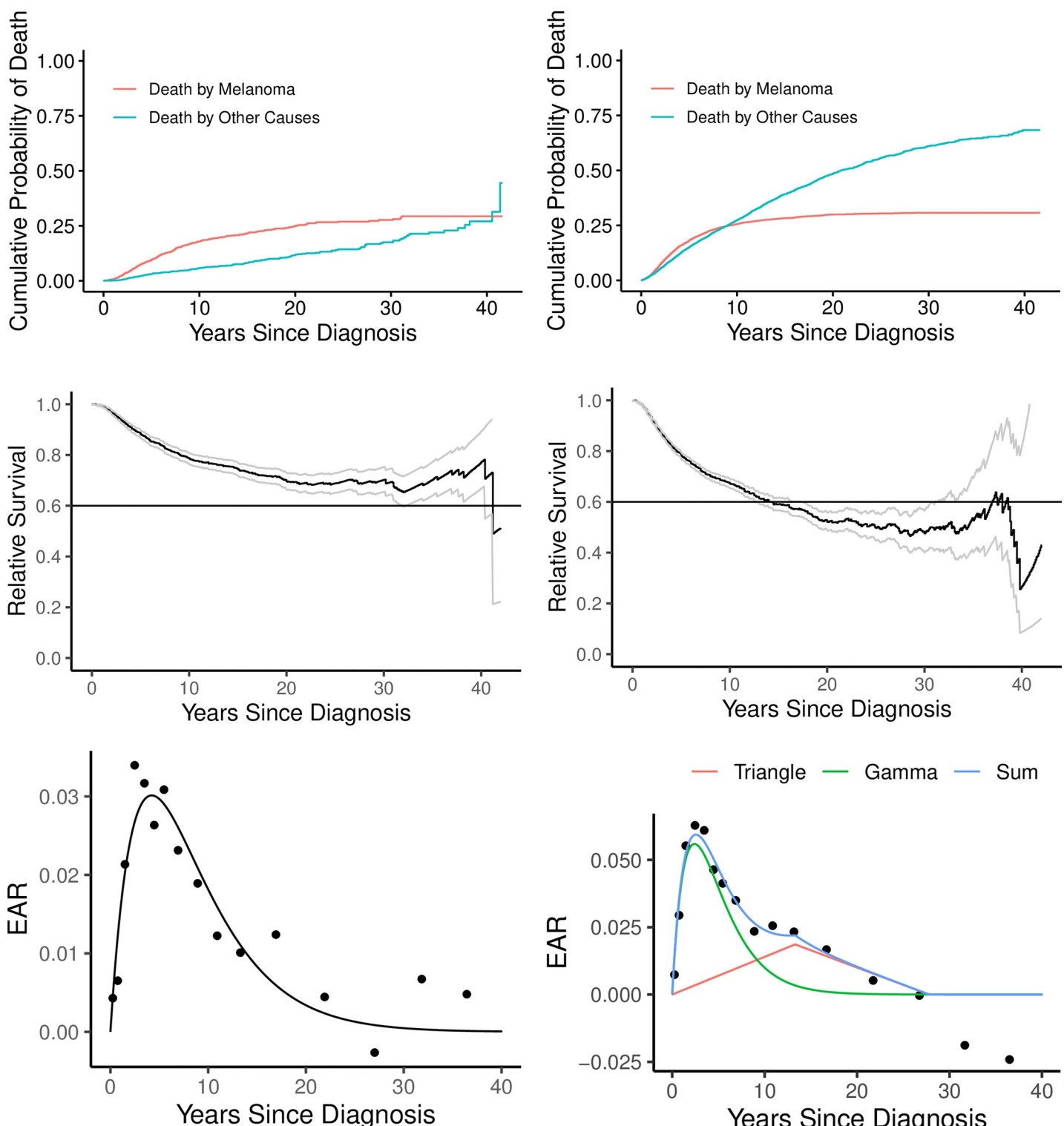

**Fig 3. Dependence on age.** Those diagnosed before the age of 50 years are shown on the left (A, C, E) and those diagnosed at ≥50 years are shown on the right (B, D, F). There are fewer competing risks in younger vs older patients (A vs B), so the cross-over time is much later. Fewer competing risks in younger patients does not, however, imply a greater overall probability of death by melanoma, as final percentage are ~30% vs ~50% (C vs D). The reason is that the disease itself is less lethal in younger patients (E vs F).

**Table 3. Uveal melanoma: Long-term survival.** Published data and methods.

| | Author, Year | Setting N | Follow up years Median (range) | Method | Year since diagnosis / treatment | | | | | Comment |
|---|---|---|---|---|---|---|---|---|---|---|
| | | | | | 15 | 20 | 25 | 30 | 35 | |
| 1 | Jensen 1981 | National (Denmark) 302 | 25 (25–25) | 1-MFS | 50 | 58 | 60 | X | X | Iris included |
| 2 | Kujala 2003 | National Finland 289 | 28 | Competing Risks | 45 | 48 | 49 | 50 | 52 | Iris excluded |
| 3 | Shields 2009 | Institutional 8033 | 2.8 (0–36.3) | 1-MFS^ | 32 | 37 | X | X | X | Iris included |
| 4 | Lane 2015 | Institutional 1490 | 12.3 (1–33.5) | 1-MFS | 25 | 26 | 26 | X | X | Iris excluded |
| 5 | Bagger 2018 | National Denmark 1637 | 5.1 (0.01–32.6) | Competing Risks | 34 | 44 | 44 | 44 | X | Iris excluded |
| 6 | Present Study 2020 | National US 10678 | 5.3 (0.02–42.6) | 1-MFS | 31 | 35 | 37 | 38 | 39 | Iris included |
| | | | | Competing Risks | 27 | 29 | 30 | 30 | 31 | |
| | | | | 1-RS | 35 | 40 | 40 | 40 | * | |
| | | | | EAR | 37 | 41 | 42 | 42 | 43 | |

^data shown for medium sized tumors only.

*unstable.

wherein data collection was based on National Cancer Registries [8, 24, 25]. There is some variability in the reported outcomes due to variations in the study population (iris melanoma included [6, 8] or excluded [7, 9, 25]) size of the tumors, and follow up duration [6–9, 25]. The method of ascertaining the diagnosis of metastasis as the cause of death is perhaps the biggest contributor to the variability of the reported outcomes (**Table 3**) [9].

With a median age at diagnosis of 62 years [14], it is not surprising that patients with uveal melanoma die from other causes over the long term [7]. As concluded in Danish [8] and Finnish studies [9], other causes of death are increasingly relevant over the long term [25]. In fact, in our large population-based cohort, other causes of death were more common than melanoma metastatic deaths (**Table 1**). Beyond 15 years since diagnosis, a patient with uveal melanoma was over 5 times (482 vs 85) more likely to die due to other causes than from uveal melanoma metastasis (**Fig 1**). The difference between KM-based OS and MFS increases progressively after 5 years since diagnosis, further indicating increasing burdens of other causes of death as patients age (**Fig 2A**). Accounting for competing risks, the cumulative probability of death due to other causes is greater than due to melanoma after 10 years since diagnosis (**Fig 2B**). This is also observed as a narrowing of the gap between OS and the corresponding expected survival curve after 25 years (**Fig 2C**). Moreover, uveal melanoma mediated death risks were dominant for longer duration in those younger than 50 years old at diagnosis (**Fig 3A**) than those ≥50 years old (**Fig 3B**) (~40 years and ~10 years, respectively).

Another important variable is the statistical method used to analyze the data. Mortality due to melanoma is often calculated as 1-MFS [26] which adjusts for variable follow up duration of each patient [26, 27] but considers deaths due to other causes as censored, which could introduce bias [28]. In contrast, cumulative incidence (CI) methods, as used in the COMS [2, 3] and Finnish [9] studies, treat deaths by melanoma and other causes as two competing events. Intuitively, this is more appealing. Uveal melanoma 1-MFS estimates are greater than CI estimates by as much as 25% (40% vs 50% survival) [9, 29].

Although CI estimates are more appropriate than KM estimates, they still rely on accurate cause of death recordings [9, 29]. To circumvent the need to identify the exact cause of death (metastasis or other) [5, 7, 10], relative survival has been used to report survival outcomes [11, 30]. We used SEERaBomb to merge all three SEER databases (SEER-9+SEER-13+SEER 18) to extract more cases (10000+) than can be analyzed conventionally using SEER*stat [17]. The

demographics of the present study cohort are comparable to those in an earlier SEER-based uveal melanoma study [14].

Life expectancy approaches to uveal melanoma data analysis were pioneered by Damato, Eleteri, Taktak and Coupland in the setting of uveal melanoma prognostication [28, 31]. Our approach of estimating EAR is similar. In it, individuals are pooled such that, instead of contributing times to events, they contribute person years at risk and deaths to time intervals. This converts survival analysis problems into Poisson regressions of models of numbers of deaths in each time interval. Such analyses require large amounts of data, as is provided by SEER. Our findings of modeled EAR values that return to 0 at around 25 years is consistent with limited published long-term data. Lane et al reported that annual rates of death from melanoma decreased gradually after year 6, but did not drop below 1% until 14 years after treatment [7]. In a Finnish study 90% (95% CI, 84–95) of the uveal melanoma deaths occurred within 15 years [9]. A Swedish study showed that excess mortality reached baseline values by ~15 years [4]. Force of mortality peak values of 0.06/y were also similar to those of 0.08/y in the Swedish study [4].

The estimated lifetime risk of death due to melanoma, using a variety of statistical methods, ranged from 30–43%. The lower values of 30% (95% CI: 29%-31%) and 32% were estimated by competing risk analysis and by simple calculation of raw proportions, respectively. Both these methods rely on knowing the exact cause of death. Given that these values are on the lower end of the range of our estimates and also lower than those reported in Danish (50%) [8] and Finnish (52%) [9] studies, where exact cause of death was established with meticulous details due to low number of study cases (302 and 289, respectively), under reporting of metastatic deaths in the SEER data set is the most likely explanation [32].

The estimated lifetime risk of death due to melanoma by KM analysis of 39% (95% CI: 37.1%, 41.6%) was 9 percentage point higher than that estimated by competing risk analysis (30%, 95% CI: 29%-31%), a value that is strikingly similar to one reported by Kjujala et al. As causes death in any cancer patient cannot be ascertained with high accuracy outside of a well-defined study, with competing causes of death in an aging population, the lifetime risk of death by uveal melanoma is likely 40–43% in the US. For patient counseling purposes, conditional survival, which is a dynamic estimate of survival probability with additional years survived may be more relevant particularly for patients who have survived initial 5–10 years [33]. Exploration of EAR over the long term and the time at which the overall EAR reaches 0 may be used to estimate time to cure and cured fractions in uveal melanoma [34].

In conclusion, relative survival methods can be used to estimate long term survival of uveal melanoma patients without knowing the exact cause of death. RS and EAR provide more realistic estimates as they compare the survival to that of a normal matched population. Death due to melanoma are rare beyond 20 years. The majority of uveal melanoma patients will die from unrelated causes and those still alive will reach a normal life expectancy about 25 years after ocular therapy.

## Supporting information

**S1 Text. R scripts used in "uveal melanoma: Long-term survival".**
(PDF)

## Author Contributions

**Data curation:** Tomas Radivoyevitch.

**Formal analysis:** Tomas Radivoyevitch, Emily C. Zabor.

**Methodology:** Emily C. Zabor, Arun D. Singh.

**Project administration:** Arun D. Singh.

**Supervision:** Arun D. Singh.

**Writing – original draft:** Tomas Radivoyevitch, Emily C. Zabor, Arun D. Singh.

**Writing – review & editing:** Tomas Radivoyevitch, Emily C. Zabor, Arun D. Singh.

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
