## [Decision Letter · Decision Letter 0]

13 Apr 2021

PONE-D-21-05440

Uveal Melanoma : Long-Term Survival

PLOS ONE

Dear Dr. Singh,

Thank you for submitting your manuscript to PLOS ONE. After careful consideration, we feel that it has merit but does not fully meet PLOS ONE’s publication criteria as it currently stands. Therefore, we invite you to submit a revised version of the manuscript that addresses the points raised during the review process.

The authors present ways of estimating the probability of long-term survival in uveal melanoma patients in US using statistical methods on data from SEER databases. This is an important question to answer. There are some concerns by the reviewers regarding methodology and discussion which can hopefully be addressed. 

We look forward to receiving your revised manuscript.

Kind regards,

Pukhraj Rishi

Academic Editor

PLOS ONE

Additional Editor Comments:

This is an interesting article on ways of estimating the probability of long-term survival in uveal melanoma patients in US using statistical methods on data from SEER databases. This is an important question to answer. There are some concerns by the reviewers regarding methodology and discussion which can hopefully be addressed.

Journal Requirements:

3. Please include captions for all your Supporting Information files at the end of your manuscript, and update any in-text citations to match accordingly. Please see our Supporting Information guidelines for more information: http://journals.plos.org/plosone/s/supporting-information.

 [NO].

[NO].

Reviewers' comments:

Reviewer's Responses to Questions

**Comments to the Author**

1. Is the manuscript technically sound, and do the data support the conclusions?

Reviewer #1: Yes

Reviewer #2: Yes

2. Has the statistical analysis been performed appropriately and rigorously? 

Reviewer #1: Yes

Reviewer #2: I Don't Know

3. Have the authors made all data underlying the findings in their manuscript fully available?

Reviewer #1: Yes

Reviewer #2: Yes

4. Is the manuscript presented in an intelligible fashion and written in standard English?

Reviewer #1: Yes

Reviewer #2: Yes

5. Review Comments to the Author

Reviewer #1: 1. The authors present ways of estimating the probability of long-term survival in uveal melanoma patients using statistical methods on data from SEER databases.

2.The data comprises all patients diagnosed as uveal melanoma between 1976 and 2016. The authors report that 5625 of the 10678 patients are alive as of 12/31/2016. Effectively the follow up data available is restricted to the end of year 2016 which limits the follow up period of the patients diagnosed with melanoma in the later part of this cohort. In fact, for the patients diagnosed in the year 2016, there is almost no follow up data.

This is also borne out from the table 3 where they indicate that the mean follow up in this study was 5.3 yrs (range of 0.02 to 42.6). The National Denmark study in contrast had a mean follow up of 25 yrs although the total number of cases in that study was far less.

Intuitively one would feel that including subjects with no follow up would affect the model being constructed to predict the long-term overall survival rates.

The authors may consider restricting the analysis to patients with a minimum of 5 yrs follow up to enable better inferences to be drawn.

3. Since very few patients die of melanoma after 10 yrs follow up. Can one sub analyse the age groups less than 50 Vs more than 50 whether this statement holds true for all age groups. In other words, is it possible that in the older population, the concurrent natural causes of death overtake the melanoma in causing death (after 10yrs) while in younger population, it is possible that melanoma may still be important even after 10 yrs in the causation of death.

Reviewer #2: This is a nice manuscript on an important and poorly investigated topic of the long term survival of uveal melanoma patients. As the authors point out this is a difficult subject to evaluate.

As this is mostly a statistical paper in terms of methodology, it is really very dense and difficult to read. I realize that the information needs to be displayed but it may be worthwhile to use % in areas whenever possible (instead of presenting the percentages as 0.xx) for readability and to include more summative statements and tables. right now I suspect that the vast majority of clinicians, who the paper is directed to, will stop reading this in its current version.

Even the abstract and discussion require some more clear, declarative summary statements than in this current version.

Some other considerations for statements that will be impactful for clinicians and patients:

risk of melanoma related death per year after diagnosis of melanoma. Risk of non-melanoma related death per year.

risk in first 5 years, 10 years

risk that decreases after 5 years in terms of melanoma specific mortality

basically, please try to add some clear, clinically relative take home statements in a clear and readable fashion

Figure 2a and 2b are invert of the same information, if space is needed one could be removed

I am not sure if Figure 3 adds significantly to Figure 2

6. PLOS authors have the option to publish the peer review history of their article (what does this mean?). If published, this will include your full peer review and any attached files.

Reviewer #1: No

Reviewer #2: No

---

## [Author Response · Author response to Decision Letter 0]

14 Apr 2021

Editor Comments:

This is an interesting article on ways of estimating the probability of long-term survival in uveal melanoma patients in US using statistical methods on data from SEER databases. This is an important question to answer. There are some concerns by the reviewers regarding methodology and discussion which can hopefully be addressed.

Thank you! Reviewer concerns are addressed point-by-point below.

Journal Requirements:

1. Please ensure that your manuscript meets PLOS ONE's style requirements, including those for file naming. The PLOS ONE style templates can be found at …

The manuscript is now formatted for PLOS ONE. 

The data is publicly available via the NCI’s SEER program upon signing a Data Use Agreement with the NCI. Anyone accessing this data at the individual level must sign this agreement. Thus, regarding your link above, we are not restricting access for any reasons stated therein. 

Grouped data used in the EAR Poisson regressions of Figure 2E (formerly Fig S1) is now available via a GitHub link provided at the end of S1 Text. 

3. Please include captions for all your Supporting Information files at the end of your manuscript, and update any in-text citations to match accordingly. Please see our Supporting Information guidelines for more information: http://journals.plos.org/plosone/s/supporting-information.

We now follow the Supporting Information formatting guidelines described in the link above. 

 [NO].

 Please clarify the sources of funding (financial or material support) for your study. List the grants or organizations that supported your study, including funding received from your institution.

 State what role the funders took in the study. If the funders had no role in your study, please state: “The funders had no role in study design, data collection and analysis, decision to publish, or preparation of the manuscript.”

 If any authors received a salary from any of your funders, please state which authors and which funders.

 If you did not receive any funding for this study, please state: “The authors received no specific funding for this work.” 

There is no funding associated with this work. 

[NO].

We have no competing interests.

Comments to the Author

1. Is the manuscript technically sound, and do the data support the conclusions?

Reviewer #1: Yes

Reviewer #2: Yes

Thank you!

2. Has the statistical analysis been performed appropriately and rigorously? 

Reviewer #1: Yes

Reviewer #2: I Don't Know

Thank you!

3. Have the authors made all data underlying the findings in their manuscript fully available?

Reviewer #1: Yes

Reviewer #2: Yes

See response above regarding our inclusion of grouped data underlying Fig. 2E.

4. Is the manuscript presented in an intelligible fashion and written in standard English?

Reviewer #1: Yes

Reviewer #2: Yes

Thank you!

5. Review Comments to the Author

Reviewer #1: 

 The authors present ways of estimating the probability of long-term survival in uveal melanoma patients using statistical methods on data from SEER databases. The data comprises all patients diagnosed as uveal melanoma between 1976 and 2016. The authors report that 5625 of the 10678 patients are alive as of 12/31/2016. Effectively the follow up data available is restricted to the end of year 2016 which limits the follow up period of the patients diagnosed with melanoma in the later part of this cohort. In fact, for the patients diagnosed in the year 2016, there is almost no follow up data.

This is also borne out from the table 3 where they indicate that the mean follow up in this study was 5.3 yrs (range of 0.02 to 42.6). The National Denmark study in contrast had a mean follow up of 25 yrs although the total number of cases in that study was far less. Intuitively one would feel that including subjects with no follow up would affect the model being constructed to predict the long-term overall survival rates.

The authors may consider restricting the analysis to patients with a minimum of 5 yrs follow up to enable better inferences to be drawn.

We appreciate this point as it is important in crude estimates of percentage of deaths caused by uveal melanoma. Thus, the first paragraph in our Results section provides this estimate focusing on SEER-9 diagnoses in 1975-1986 (i.e. cases with at least 30 years of follow up). “ Focusing on SEER-9 diagnoses in 1975-1986, 37% [399/(399+678)] were due to uveal melanoma. Assuming 163 still alive at the end of 2016 will die of other causes, the proportion falls to 32%. This raw proportion is our first estimate of the lifetime risk of death due to uveal melanoma.” For hazard function-based results in Fig. 2, however, we do not expect this to be an issue. 

 Since very few patients die of melanoma after 10 yrs follow up. Can one sub analyse the age groups less than 50 Vs more than 50 whether this statement holds true for all age groups. In other words, is it possible that in the older population, the concurrent natural causes of death overtake the melanoma in causing death (after 10yrs) while in younger population, it is possible that melanoma may still be important even after 10 yrs in the causation of death.

Thank you, this is a very interesting idea. 

Added new analysis in Results, new Figure 3 and also in discussion .

Age Dependence

Uveal melanoma mediated death risks are dominant for ~40 years in those younger than 50 years old at diagnosis (Figure 3A), but only for ~10 years in those ≥50 years old (Figure 3B). Lifetime risks of death by melanoma in these two age groups are ~30% (Figure 3C) and ~50% (Figure 3D), respectively, based on relative survival. Furthermore, in the younger age group (Figure 3E) relative to the older age group (Figure 3F), the peak EAR is lower and the triangle wave is not detectable. Negative EARs at large times in Figure 3F are possible, if additional surveillance results in better health care, but 95% confidence intervals of these estimates are wide and include 0, so we cannot claim this: the estimates of the last two EARs are -0.0188 (-0.0598, 0.0222) and -0.0241 (-0.113, 0.0648). 

Reviewer #2: 

This is a nice manuscript on an important and poorly investigated topic of the long term survival of uveal melanoma patients. As the authors point out this is a difficult subject to evaluate.

As this is mostly a statistical paper in terms of methodology, it is really very dense and difficult to read. I realize that the information needs to be displayed but it may be worthwhile to use % in areas whenever possible (instead of presenting the percentages as 0.xx) for readability and to include more summative statements and tables. right now I suspect that the vast majority of clinicians, who the paper is directed to, will stop reading this in its current version.

Even the abstract and discussion require some more clear, declarative summary statements than in this current version.

Thank you for this feedback. Fractions were changed to percentages to improve readability.

Some other considerations for statements that will be impactful for clinicians and patients:

risk of melanoma related death per year after diagnosis of melanoma. Risk of non-melanoma related death per year. risk in first 5 years, 10 yearsrisk that decreases after 5 years in terms of melanoma specific mortality

Thank you for these suggestions. We believe that required info is covered in Figure 1 displayed as bar chart and in Table 2. Made it explicit by modifying the legend and added in results and abstract. 

“In the first 5 years since diagnosis of uveal melanoma, the proportion of deaths attributable to uveal melanoma were 1.3 with rapid fall after 10 years. Death due to melanoma are rare beyond 20 years (Figure 1).”

The reviewer is correct in pointing out about study density of the paper as the emphasis is on the methodology and variations in the estimates based upon the method used.

Survival stats at 5 and 10 years after diagnosis were calculated, conditioned on various numbers of years already survived is in other published work by us. Added in discussion

“For patient counseling purposes, conditional survival, which is a dynamic estimate of survival probability with additional years survived may be more relevant particularly for patients who have survived initial 5-10 years.[33] Exploration of EAR over the long term and the time at which the overall EAR reaches 0 may be used to estimate time to cure and cured fractions in uveal melanoma [34].” 

Reference 33. Zabor, EC et al. Conditional Survival in Uveal Melanoma, Ophthalmol Retina. 2020 Sep 23;S2468-6530(20)30395-X. doi: 10.1016/j.oret.2020.09.015. 

Reference 34. Singh et al . Cured fractions in uveal melanoma . JAMA Ophthal 2021

Figure 2a and 2b are invert of the same information, if space is needed one could be removed

Not really. In 2A death by other causes equates to loss of follow up and in 2B it is on an equal footing as death by uveal melanoma. Please note that there is cross-over in 2B not present in 2A. 

I am not sure if Figure 3 adds significantly to Figure 2

Agreed. As it is a summary of Fig. 2, it is now Fig 2F (Figure S1 is now Fig. 2E).

---

## [Editor Report · Decision Letter 1]

19 Apr 2021

Uveal Melanoma : Long-Term Survival

PONE-D-21-05440R1

Dear Dr.  Singh,

We’re pleased to inform you that your manuscript has been judged scientifically suitable for publication and will be formally accepted for publication once it meets all outstanding technical requirements.

Kind regards,

Pukhraj Rishi

Academic Editor

PLOS ONE
---

## [Editor Report · Acceptance letter]

28 Apr 2021

PONE-D-21-05440R1 

Uveal Melanoma : Long-Term Survival 

Dear Dr. Singh:

I'm pleased to inform you that your manuscript has been deemed suitable for publication in PLOS ONE. Congratulations! Your manuscript is now with our production department. 

Kind regards, 

on behalf of

Dr. Pukhraj Rishi 

Academic Editor

PLOS ONE